Accessibility challenges of e-commerce websites

http://orcid.org/0000-0003-4210-0117 Acosta-Vargas Patricia 1 patricia.acosta@udla.edu.ec
Salvador-Acosta Belén 2
Salvador-Ullauri Luis 3
http://orcid.org/0000-0002-3616-2074 Jadán-Guerrero Janio 4
1 Intelligent and Interactive Systems Laboratory/FICA/Industrial Engineering, Universidad de Las Américas - Ecuador , Quito , Ecuador
2 Facultad de Medicina, Universidad de Las Américas - Ecuador , Quito , Ecuador
3 Department of Software and Computing Systems, University of Alicante , Alicante , España
4 Centro de Investigación en Mecatrónica y Sistemas Interactivos - MIST, Universidad Tecnológica Indoamérica , Quito , Ecuador
Gupta Varun
Electronic publication date: 2022 Feb 22
Publication date: 2022
Volume: 8
Electronic Location ID: e891
Received 2021 Sep 8; Accepted 2022 Jan 24
Copyright: © 2022 Acosta-Vargas et al.
Copyright year: 2022
Copyright holder: Acosta-Vargas et al.
License: This is an open access article distributed under the terms of the Creative Commons Attribution License, which permits unrestricted use, distribution, reproduction and adaptation in any medium and for any purpose provided that it is properly attributed. For attribution, the original author(s), title, publication source (PeerJ Computer Science) and either DOI or URL of the article must be cited.
License URL: https://creativecommons.org/licenses/by/4.0/

Keywords: Accessibility, e-commerce, Websites, WCAG, Web

Funding: Universidad de Las Américas-Ecuador INI.PAV.20.01 This research was funded by Universidad de Las Américas-Ecuador as an internal research project INI.PAV.20.01. The funders had no role in study design, data collection and analysis, decision to publish, or preparation of the manuscript.

==============================
Today, there are many e-commerce websites, but not all of them are accessible. Accessibility is a crucial element that can make a difference and determine the success or failure of a digital business. The study was applied to 50 e-commerce sites in the top rankings according to the classification proposed by ecommerceDB. In evaluating the web accessibility of e-commerce sites, we applied an automatic review method based on a modification of Website Accessibility Conformance Evaluation Methodology (WCAG-EM) 1.0. To evaluate accessibility, we used Web Accessibility Evaluation Tool (WAVE) with the extension for Google Chrome, which helps verify password-protected, locally stored, or highly dynamic pages. The study found that the correlation between the ranking of e-commerce websites and accessibility barriers is 0.329, indicating that the correlation is low positive according to Spearman’s Rho. According to the WAVE analysis, the research results reveal that the top 10 most accessible websites are Sainsbury’s Supermarkets, Walmart, Target Corporation, Macy’s, IKEA, H&M Hennes, Chewy, Kroger, QVC, and Nike. The most significant number of accessibility barriers relate to contrast errors that must be corrected for e-commerce websites to reach an acceptable level of accessibility. The most neglected accessibility principle is perceivable, representing 83.1%, followed by operable with 13.7%, in third place is robust with 1.7% and finally understandable with 1.5%. Future work suggests constructing a software tool that includes artificial intelligence algorithms that help the software identify accessibility barriers.

Introduction

Internet technology has radically revolutionized the world of communications to become a global means of communication. The number of e-commerce websites has increased significantly due to the pandemic COVID-19 (World Health Organization, 2021); the global confinement caused many businesses to close.

Statistics from the Statista website (Statista, 2021) indicate that e-commerce has undergone a substantial transformation in recent years thanks to digitization in modern life.

However, for Statista (2020), e-commerce websites have seen a notable increase in worldwide traffic flow between January 2019 and June 2020. By 2022, more than 2.14 billion people worldwide are forecast to shop online, and global e-commerce revenues could grow to $5.4 trillion (Statista, 2021).

Figure 1 shows the search of terms performed in the last 5 years in Google Trends (Google, 2021) related to accessibility, e-commerce, and Web Content Accessibility Guidelines (WCAG) (World Wide Web Consortium, 2018). We observe that the term e-commerce tends to grow from March 2020 during the COVID-19 where most users began to consume massively digital material and are oriented to use e-commerce applications, to reduce the number of infections, we also observe that the term accessibility and WCAG tend to grow.

Figure 1 The trend in Google Trends for terms related to accessibility, e-commerce, and WCAG.

Diagram of the trend of the terms in the last 5 years worldwide. Data source: Google Trends (https://www.google.com/trends).

E-commerce websites have grown considerably, but most of them are not accessible. Accessibility (Acosta-Vargas, Acosta & Lujan-Mora, 2018) refers to a set of techniques, guidelines or methods that make web content and functionality compatible with the needs of all people regardless of their physical or technological capabilities.

Statements from the World Health Organization (2017) reveal that 15% of the world’s population suffers from some disability. Therefore, it is essential to apply web accessibility guidelines to e-commerce sites. A well-designed website can be easy to navigate for web users (World Wide Web Consortium, 2018). Accessibility also benefits several users (Andersen, Hoss & Bridge, 2020) with aging-related difficulties that decrease their visual ability due to presbyopia.

WCAG 2.1 (World Wide Web Consortium, 2018) proposes applying accessibility principles to reduce accessibility barriers perceived by users when interrelating with a website.

The outcomes of this investigation evidenced that 25.6% of the e-commerce websites in the sample present images that require the inclusion of alternative text; additionally, we found that 54.4% of the sites present contrast problems related to the principle of perception.

As future work, we suggest considering the problems of hardware limitations and interstitial advertising. We also recommend building a software tool with artificial intelligence algorithms that include new heuristics to help developers identify accessibility barriers to generate more accessible and inclusive sites.

The remainder of this paper is structured as follows; “Literature Review” reviews the literature on web accessibility. “Web Accessibility Principles” shows the methodology used to evaluate accessibility in e-commerce sites. “Materials and Methods” shows the results obtained by applying the evaluation. “Results” contains the discussion of the outcomes. “Discussion” explains the restrictions of the research. Finally, conclusions and future work are described in parts VII and VIII.

Literature Review

The pandemic status related to COVID-19 has accelerated the movement of industry, education and business to the virtual world, and on par with these events, several e-commerce websites have been created (Munkova et al., 2021).

According to Villa & Monzón (2021), Pollák, Konečný & Ščeulovs (2021) and Paștiu et al. (2020), COVID-19 has impacted the growth of e-commerce websites; unprecedented worldwide changes are evident in the various forms of consumer habits. Consumer behavior shows an evolutionary shift from offline to online, where e-commerce applications require (1) more accessible designs, (2) greater sustainability, (3) software applications that utilize business intelligence.

In reviewing the literature, we found some research related to the evaluation of web accessibility in e-commerce sites, methods, and tools used in automatic inspection. Paz et al. (2021) argue the accessibility with which software products, including e-commerce stores, should be designed. It indicates that some countries apply laws and government policies that ensure accessibility to websites considering different skills and abilities. The study compared the results of five tools for inspecting the accessibility of e-commerce websites; the conclusions show that there are no 100% accessible sites.

Xu (2020) compares the accessibility of e-commerce, considering compliance with web accessibility guidelines. As a case study, they applied the evaluation to 45 e-commerce websites. They used the Web Accessibility Assessment Tool (WAVE); the results revealed that websites with Accessible Rich Internet Applications (ARIA) attribute lower accessibility levels overall. They concluded that the accessibility of mature websites was higher than that of new websites with innovative products.

Alshamari (2016) argues that many tools can help make a website accessible. The article explores some available tools that help designers and developers evaluate web accessibility. The research results indicate that navigation, readability, and timing are the most common accessibility issues when evaluating the accessibility of selected websites.

Padure & Pribeanu (2020) argues that WAVE tool is a free tool provided by Web Accessibility In Mind (WebAIM). The authors indicate that WAVE offers a color-coding system: red for errors that need to be corrected urgently, green for correct lines but still need to be checked, and yellow for potential problems that need manual review.

Oliveira, Afonso & Pinto (2020) infer that accessibility is fundamental in the democratization of technologies, so applying the Web Content Accessibility Guidelines (WCAG 2.1) is essential. The accessibility evaluation was applied to three websites operating in Portugal, considering the three best-positioned retailers’ ranking corresponding to the SimilarWeb. The results obtained established a collection of suggestions to increase the accessibility of websites aimed at e-commerce. Our study differs from Oliveira, Afonso & Pinto (2020) and Xu (2020) because the total sample is taken from ecommerceDB, which presents the e-commerce websites related to market trends and a ranking of the leading e-commerce stores. Our evaluation applied a new method based on the methodology (WCAG-EM) 1.0. In addition, the WAVE evaluation tool is based on version 3.1.6, updated as of October 14, 2021, which includes the plugin component that allows evaluating websites that require authentication. We, therefore, propose 10 recommendations to improve the accessibility of the websites listed in the discussion section.

Research (Abascal, Arrue & Valencia, 2019) related to Web accessibility evaluation argues that manual verification of compliance with accessibility guidelines is often complicated and unmanageable, so the authors suggest applying software tools that perform automatic accessibility evaluations. It presents a review of the main features of tools used for Web accessibility evaluation and presents an introspection of the future of accessibility tools.

Acosta-Vargas, Antonio Salvador-Ullauri & Lujan-Mora (2019) suggest that verifying the accessibility of a Web site is a considerable challenge for accessibility specialists. Today, there are quantitative and qualitative methods for verifying whether a website is accessible. In general, the methods use automatic tools because they are low-cost, but they do not represent a perfect solution. The authors propose a heuristic method with a manual review supported by the Web Content Accessibility Guidelines 2.1. The evaluators concluded that the research could serve as a preliminary argument for upcoming analyses concerned with web accessibility heuristics.

Our investigation proposes an automatic review method using the WAVE Web Accessibility Evaluation Tool (WebAIM, 2021). Earlier studies by Acosta-Vargas, Acosta & Lujan-Mora (2018) indicated that one of the best tools for automated review is WAVE, which allows you to identify any accessibility barriers, centered on the Web Content Accessibility Guidelines (WCAG) 2.1 (World Wide Web Consortium, 2018) that help in automatic review and evaluation by web content experts.

This preliminary study was applied to the 50 best-ranked e-commerce stores according to the ranking proposed by the ecommerceDB site (EcommerceDB, 2020), which contains detailed information on more than 20,000 stores from 50 countries and 13 categories.

Web Accessibility Principles

Previous studies indicate the increase of e-commerce in COVID-19 time; this accelerated growth has generated millions of e-commerce websites, but many sites are not accessible; therefore, it is essential to bear in mind the Web Content Accessibility Guidelines 2.1 (WCAG 2.1) proposed by the World Wide Web Consortium.

Web accessibility implies that people with incapacities can use the web. This process implies that they can recognize, identify, navigate, and relate to the website. WCAG 2.1 (World Wide Web Consortium, 2018) consists of four principles, 13 guidelines and 78 conformance such as compliance or success criteria, plus some techniques.

Principle 1: Perceptible refers to the website’s contents and the interface design for all users. It includes the audiovisual contents, the interface, images, buttons, video players and other components that must be accessible, recognizable, and feasible by any individual in any condition, tool, and operating system.

Principle 2: Operable, this means that a website should be as intuitive as possible, with options to perform an action or search for content. The more alternatives included in the site’s navigation, the better its accessibility. In other words, the website must ensure all keyboard-based functionality and avoid designs that may cause epileptic seizures.

Principle 3: Understandable implies that the site includes legible and understandable elements, both in the form and substance of the texts. It must contain fonts that all users can read in its form. In addition, it should be predictable as to how the site works so that potential users do not waste time trying to guess how a tool works for better navigation.

Principle 4: Robust refers to websites or applications that must be compatible with all browsers, operating systems, and devices, as well as assistive technology applications or digital ramps.

The four principles of accessibility contain 13 guidelines that support the goals of website designs to make content more accessible to users with disabilities. Each guideline includes success criteria that can be applied in situations for compliance testing in a contractual agreement. To meet accessibility requirements, three conformance levels are defined, with A being the lowest, AA being a medium level and acceptable by WCAG, and AAA being the highest (World Wide Web Consortium, 2018).

The documentation includes several techniques; the techniques are grouped into two classes (1) those that are sufficient to achieve the success criteria and (2) advisable techniques that allow the authors to better comply with the guidelines. In addition, some of the advisable techniques deal with accessibility barriers that the verifiable success criteria have not covered.

Materials and Methods

In this research to evaluate the accessibility of e-commerce websites, we applied an automatic review method (World Wide Web Consortium, 2014) centered on a modification of the Website Accessibility Conformance Evaluation Methodology (WCAG-EM) 1.0.

We used Web Accessibility Evaluation Tool (WAVE) (WebAIM, 2021) with the extension for Google Chrome, which helps verify password-protected and highly dynamic pages. Utah State University developed the WAVE automatic evaluation tool to help find potential accessibility issues according to WCAG 2.1, facilitating manual evaluation. It should be noted that manual testing cannot be replaced, especially when it comes to accessibility, as it may be essential to test with end-users with disabilities. Accessibility validation was performed using guidelines based on Section 508 and WCAG 2.1; some phases of this methodology (Salvador-Ullauri et al., 2020) were tested in previous works of the authors related to serious games; the methodology for evaluating e-commerce websites is summarized in eight phases, as shown in Fig. 2.

Figure 2 Methodology for evaluating e-commerce websites.

Diagram for assessing accessibility in e-commerce websites.

Phase 1 select e-commerce websites

In this phase, we selected the 50 e-commerce websites that are in the top rankings according to the classification proposed by ecommerceDB (EcommerceDB, 2020). In this preliminary phase, we define the level of compliance with WCAG 2.1 (World Wide Web Consortium, 2018). In this case, we evaluated the AA level within the level accepted and recommended by WCAG 2.1. This knowledge can be improved to propose future work; we also determined the lowest configuration of web browser patterns, operating systems and assistive technologies with which the website should work during the testing phase. This case study used the Windows 10 operating system with Google Chrome browser version 95.0.4638.54 and screen reader support.

Phase 2 categorize the type of users

According to Salvador-Ullauri et al. (2020), in this phase, we involved three web accessibility experts with experience in the area since 2015, who have more than 10 scientific publications in web accessibility evaluations, serious games, and accessible mobile applications. Discrepancies found in the automatic review of e-commerce sites were resolved in consensus. Experts performed the automatic review with WAVE; as evaluated by experts, WAVE is one of the best performing tools according to previous studies (Acosta-Vargas, Acosta & Lujan-Mora, 2018). This phase identified the flow of events users interact with when browsing e-commerce sites.

Phase 3 define the test scenario

According to Salvador-Ullauri et al. (2020), this phase identifies the essential functionalities of the e-commerce website to help select the most representative instances. The definition of the test scenario serves as the basis for the subsequent selection of the e-commerce sites. In this case, we apply the following scenario: (1) We enter the first page of the website. (2) We interact by selecting and purchasing products on the e-commerce website. (3) We test how to fill out and submit the forms. (4) We check if account registration on the e-commerce site is required.

Phase 4 explore the e-commerce website

In this phase, the first page of each website was explored. The evaluators explored the e-commerce website to understand its purpose, functionality, and usage. Initial exploration of this phase was considered in Phase 1 by selecting a representative sample, then refined in phase 5 by evaluating with WAVE. Involving accessibility experts and website designers can help get the scans more efficiently. At first, cursory checks were performed to help identify relevant web pages; later, a more detailed evaluation of each website component was performed. Therefore, this phase is essential for evaluators to access all the essential components and functionalities of the website.

Phase 5 evaluate with WAVE

In this phase, the experts used WAVE to evaluate the home page of each e-commerce website. The assessment was conducted in March 2021; during this phase, the evaluators audited the sample e-commerce websites and the states of the websites selected in phases 1 and 4.

The evaluation was conducted following the WCAG 2.1 conformance requirements at the AA level previously defined in phase 1. The conformance level, web pages, processes and technologies, and compatibility with accessibility and non-interference were considered. In this phase, it was essential to know the WCAG 2.1 (World Wide Web Consortium, 2018) conformance requirements and the experience of accessibility experts. In addition, the authors classified accessibility barriers by matching the WCAG 2.1 principles, guidelines, and success criteria, which were then validated with WAVE results.

Phase 6 record evaluation data

In this phase, the assessment data was documented in a spreadsheet to organize a dataset available in Mendeley (Acosta-Vargas et al., 2021). The dataset includes information with the names of the audited websites, the URLs of the e-commerce sites, and the evaluation data used to replicate this study as part of good practices that help researchers. The set organizes the information into spreadsheets, containing (1) The e-commerce websites evaluated. (2) The results of the e-commerce websites evaluated with WAVE. (3) The map of the number of e-commerce websites per country. (4) The diagram of the accessibility evaluation process with WAVE. (5) A summary of the evaluation of e-commerce websites. (6) The accessibility principles of WCAG 2.1. (7) Accessibility barriers identified when evaluating with WAVE. (8) E-commerce websites vs compliance. (9) E-commerce websites with errors and contrast errors. (10) The number of alerts, features, structural elements and ARIA. (11) E-commerce websites and the relation to accessibility levels.

Phase 7 classify and analyze data

Following previous work by Salvador-Ullauri et al. (2020), data related to accessibility principles, success criteria, and accessibility levels are organized in this phase. This information is detailed in the results section and discussed in the discussion section. Also, the e-commerce websites were classified by (1) The countries to which each domain corresponds according to the registered URL. (2) The severe errors in need of correction to remove accessibility barriers. (3) Contrast errors that make access difficult for visually impaired users. (4) The ranking in which they are placed and the level of accessibility. The data analysis was performed with Microsoft Excel version 365 MSO 16.0.14326.20504, with macros, advanced functions, tables, and dynamic graphs.

Phase 8 suggest accessibility improvements

In this phase, proposals for improvements to the e-commerce websites were presented. The improvements are detailed in the discussion section.

Results

This research was applied to a sample of the top 50 e-commerce websites taken from ecommerceDB; which contains information on more than 20,000 e-shops from around 50 countries. It is divided into several categories such as revenue and competitor analysis, market development, performance and traffic indicators, vendor submission, payment options, social media activity and SEO information. The ecommerceDB.com database also covers e-commerce market analysis, customer behaviors and buying patterns, market trends and company histories. Table 1 contains the sites that were evaluated with WAVE.

Table 1 E-commerce websites.

A sample of 50 e-commerce stores according to the ranking of the classification proposed by ecommerceDB, followed by the name of the electronic store, the URL, and the acronym.

#	Online-store	URL	Acronym	
1	Amazon.com, Inc.	https://www.amazon.com/	amazon_Us	
2	Beijing Jingdong 360 Degree E-Commerce Co., Ltd.	https://global.jd.com/	global	
3	Apple, Inc.	https://www.apple.com/	apple	
4	Walmart, Inc.	https://www.walmart.com/	walmart	
5	Suning Tesco Group Co., Ltd.	https://www.suning.com/	suning	
6	Amazon EU S.à r.l.	https://www.amazon.de/	amazon_De	
7	Amazon EU S.à r.l.	https://www.amazon.co.uk/	amazon_Uk	
8	Guangzhou Vipshop Electronic Commerce Co., Ltd.	https://www.vip.com/	vip	
9	Target Corporation	https://www.target.com/	target	
10	Amazon.com Services, LLC	https://www.amazon.co.jp/	amazon_Jp	
11	Guangdong Midea Network Technology Co., Ltd.	https://www.midea.cn/	midea	
12	The Home Depot, Inc.	https://www.homedepot.com/	homedepot	
13	Best Buy Co., Inc.	https://www.bestbuy.com/	bestbuy	
14	Amazon.com.ca, Inc.	https://www.amazon.ca/	amazon_Ca	
15	Wayfair, LLC	https://www.wayfair.com/	wayfair	
16	Amazon EU S.à r.l.	https://www.amazon.fr/	amazon_Fr	
17	Costco Wholesale Corporation	https://www.costco.com/	costco	
18	Chewy, Inc.	https://www.chewy.com/	chewy	
19	Nike, Inc.	https://www.nike.com/xl/	nike	
20	Inter IKEA Systems B.V.	https://www.ikea.com/	ikea	
21	Tesco Stores, Ltd.	https://www.tesco.com/	tesco	
22	Macy’s, Inc.	https://www.macys.com/	macys	
23	Argos, Ltd.	https://www.argos.co.uk/	argos	
24	Wildberries, OOO	https://www.wildberries.ru/	wildberries	
25	Huawei Device Co., Ltd.	https://www.vmall.com/	vmall	
26	Kohl’s Corporation	https://www.kohls.com/	kohls	
27	Amazon EU S.à r.l.	https://www.amazon.es/	amazon_Es	
28	W.W. Grainger, Inc.	https://www.grainger.com/	grainger	
29	H&M Hennes & Mauritz GBC AB	https://www.hm.com	hm	
30	Sam's West, Inc.	https://www.samsclub.com/	samsclub	
31	Amazon EU S.à r.l.	https://www.amazon.it/	amazon_It	
32	Otto GmbH & Co. KG	https://www.otto.de/	otto	
33	The Kroger, Co.	https://www.kroger.com/	kroger	
34	HQG, Ltd.	https://www.kaola.com/	kaola	
35	Zara USA, Inc.	https://www.zara.com/	zara	
36	QVC, Inc.	https://www.qvc.com/	qvc	
37	Dell, Inc.	http://www1.la.dell.com/	la	
38	Lowe’s Home Centers, LLC	https://www.lowes.com/	lowes	
39	asos.com, Ltd.	https://www.asos.com/	asos	
40	Sainsbury’s Supermarkets, Ltd.	https://www.sainsburys.co.uk/	sainsburys	
41	Nordstrom, Inc.	https://www.nordstrom.com/	nordstrom	
42	DSG Retail, Ltd.	https://www.currys.co.uk/	currys	
43	Gap, Inc.	https://www.gap.com/	gap	
44	МVM, OOO	https://www.mvideo.ru/	mvideo	
45	Auchan Retail France SAS	https://www.auchan.fr/	auchan	
46	ZoeTop Business Co., Ltd.	https://www.shein.com/	shein	
47	John Lewis Plc	https://www.johnlewis.com/	johnlewis	
48	Magazine Luiza S.A.	https://www.magazineluiza.com.br/	magazineluiza	
49	Ocado Retail, Ltd.	https://accounts.ocado.com/	ocado	
50	Newegg, Inc.	https://www.newegg.com/	newegg	

The evaluation of the accessibility of e-commerce stores was carried out with the WAVE automatic review tool. Rich Internet applications tend to dynamically update the Document Object Model (DOM) structure, which is why the method used by WAVE to analyze the rendered DOM of pages uses heuristics and logic to detect end-user accessibility barriers considering WCAG 2.1 (World Wide Web Consortium, 2018). All automatic review tools, including WAVE, have limitations; they can detect barriers in 35% of possible compliance failures (WebAIM, 2021). The method applied in evaluating the accessibility of e-commerce stores was based on a modification of the World Wide Web Consortium (2014) WCAG-EM 1.0; our method consists of an eight-phase process detailed in Fig. 2.

Table 2 presents the data obtained from the accessibility evaluation of e-commerce websites with WAVE. It contains the number of WCAG 2.1 compliance failures that may affect specific users. Web developers should correct the barriers identified in the evaluation to make the e-commerce site accessible and inclusive. The contrast errors found in this study are related to the text that violates WCAG 2.1 contrast requirements. The term alerts are related to elements that may cause accessibility problems; in this case, the evaluator is the one who decides the impact of the accessibility of the website. Features imply that elements can improve accessibility when implemented correctly. Structural elements are related to some title of a web page, indicating that it has been marked as a top-level title or related to several milestones. Finally, the ARIA element presents information about accessibility for people with disabilities; in such a way, it influences accessibility when misused.

Table 2 E-commerce websites evaluated.

It presents a sample of 50 e-commerce stores according to the ranking of the classification proposed by ecommerceDB, followed by the acronym, errors, contrast errors, alerts, features, structural elements, ARIA, and the country to which each e-commerce corresponds.

Ranking	Acronym	Errors	Contrast
errors	Alerts	Features	Structural elements	ARIA	Country	
1	amazon_Us	4	4	106	140	46	270	United States	
2	global	24	35	47	11	13	2	Greater China	
3	apple	10	21	17	24	61	122	United States	
4	walmart	1	6	72	104	20	321	United States	
5	suning	102	34	113	112	29	3	Greater China	
6	amazon_De	4	4	94	217	56	380	Germany	
7	amazon_Uk	8	2	97	189	56	387	United Kingdom	
8	vip	73	53	28	12	22	1	Greater China	
9	target	1	6	11	45	31	125	United States	
10	amazon_Jp	5	1	32	37	31	104	Japan	
11	midea	21	83	57	222	59	0	Greater China	
12	homedepot	26	40	93	86	108	247	United States	
13	bestbuy	6	0	4	19	19	2	United States	
14	amazon_Ca	6	1	142	218	56	572	Canada	
15	wayfair	41	0	14	66	279	329	United States	
16	amazon_Fr	4	2	95	210	53	464	France	
17	costco	16	7	131	220	135	1,071	United States	
18	chewy	1	2	27	152	98	234	United States	
19	nike	2	0	105	93	28	76	United States	
20	ikea	1	6	6	18	18	120	United States	
21	tesco	9	4	47	24	131	165	United Kingdom	
22	macys	1	0	20	16	114	75	United States	
23	argos	15	2	23	14	28	195	United Kingdom	
24	wildberries	140	266	175	145	134	36	Russia	
25	vmall	107	27	29	26	47	0	Greater China	
26	kohls	38	9	287	151	25	894	United States	
27	amazon_Es	4	2	111	205	60	404	Spain	
28	grainger	9	7	34	39	86	102	United States	
29	hm	1	1	1	1	19	0	Germany	
30	samsclub	12	0	86	111	119	1,173	United States	
31	amazon_It	5	2	81	208	58	410	Italy	
32	otto	43	155	77	62	76	0	Germany	
33	kroger	2	10	50	39	44	221	United States	
34	kaola	60	246	772	246	259	2	Greater China	
35	zara	7	0	13	10	20	286	United States	
36	qvc	2	4	262	192	185	392	United States	
37	la	2	38	38	23	163	2	United States	
38	lowes	8	4	27	20	31	232	United States	
39	asos	24	3	0	66	169	366	United Kingdom	
40	sainsburys	0	6	26	53	47	47	United Kingdom	
41	nordstrom	57	6	36	11	236	49	United States	
42	currys	51	6	46	58	82	62	United Kingdom	
43	gap	11	1	23	45	33	195	United States	
44	mvideo	63	217	98	202	131	2	Russia	
45	auchan	37	233	38	61	82	57	France	
46	shein	127	33	118	72	161	618	United States	
47	johnlewis	6	17	40	27	47	39	United Kingdom	
48	magazineluiza	39	196	232	14	153	111	Brazil	
49	ocado	4	2	2	0	5	6	United Kingdom	
50	newegg	7	0	192	97	667	367	United States	

Figure 3 shows the number of e-commerce sites by country, taken as a sample for the accessibility evaluation. The most significant number of e-commerce sites evaluated corresponds to the United States, with 24 sites representing 48% of the total, followed by the United Kingdom, with eight sites representing 16%. With six e-commerce sites, Greater China accounts for 12% in third place. Next, Germany, with three sites, accounts for 6%, followed by France and Russia, with two sites each, representing 8% of the total. Lastly, Brazil, Canada, Italy, Japan and Spain, with one e-commerce site, account for 10%.

Figure 3 Map of the number of e-commerce sites taken by the country.

The map presents the countries taken as part of the sample to evaluate accessibility according to the classification proposed by ecommerceDB. The sky blue color indicates the country with the highest number of e-commerce sites, the yellow color the midpoint and the pink color the lowest number of e-commerce sites.

Figure 4 shows two categories of barriers, warning and serious barriers; the most significant warning barriers are ARIA, Structural Elements, Features and Alerts. These barriers do not affect accessibility to a high degree, and correcting them is unnecessary. The serious barriers with a high number are Contrast Errors with 1,721 barriers, representing 7.4% (pink bars) and Errors with 1,229, corresponding to 5.3% (sky blue bars), which must be corrected urgently for e-commerce sites to reach an acceptable level of accessibility. ARIA attributes (World Wide Web Consortium, 2018) add semantic information to the elements of a website, specifically for properties that help to inform: (1) The state of an element of the graphical interface. (2) The content of a section that may change when there is user interaction. (3) The elements that are part of a drag-and-drop interface. (4) The relationships between document elements.

Figure 4 Evaluation of accessibility with WAVE.

The barriers related to Errors (sky blue bars) and Contrast Errors (pink) that should be corrected urgently to improve accessibility are shown. Alerts (yellow), Features (gray), Structural Elements (orange) and ARIA (blue) can be corrected depending on the evaluator’s criteria.

Table 3 summarizes the barriers identified during the evaluation of the e-commerce sites with WAVE. Table 3 includes the barriers, success criteria, level, principle, and total barriers of the 50 e-commerce websites assessed. Table 3 comprises the success criteria composed of three numbers; the first is associated with the accessibility principle, the second to the guideline, and the third to the success criteria related to the accessibility barrier.

Table 3 Summary of the evaluation of e-commerce websites.

The summary of accessibility barriers identified by applying the WAVE automatic review tool.

Barrier	Success criteria	Level	Principle	Total	
Non-text Content	1.1.1	A	Perceivable	774	
Info and Relationships	1.3.1	A	Perceivable	94	
Contrast (Minimum)	1.4.3	AA	Perceivable	1,641	
Keyboard	2.1.1	A	Operable	8	
Bypass Blocks	2.4.1	A	Operable	12	
Link Purpose (In Context)	2.4.4	A	Operable	350	
Headings and Labels	2.4.6	AA	Operable	44	
Language of Page	3.1.1	A	Understandable	8	
Error Identification	3.3.1	A	Understandable	2	
Labels or Instructions	3.3.2	A	Understandable	35	
Name, Role, Value	4.1.2	A	Robust	51	

Figure 5 shows a synopsis of the accessibility principles recognized in the assessment of e-commerce websites. The most neglected accessibility principle is perceivable, representing 83.1% of the total, followed by operable with 13.7%, in third place is robust with 1.7%, and finally, understandable with 1.5%.

Figure 5 Accessibility evaluation using WAVE.

A detailed evaluation results with accessibility principles according to WCAG 2.1.

Figure 6 summarizes the barriers identified in the accessibility evaluation. The most affected accessibility barrier corresponds to Contrast with 54.4%, followed by Non-text Content, representing 25.6%, in third place is Link purpose, representing 11.6% of the total. The rest of the barriers, such as info and relationships, name, role, value, headings and labels, labels or instructions, bypass blocks, keyboard, the language of page and error identification, correspond to values lower than 3.1%.

Figure 6 Accessibility barriers identified when evaluating with WAVE.

The results related to the success criteria according to WCAG 2.1 are shown.

Figure 7 presents the e-commerce sites and the level of web accessibility; among the top ten most accessible websites according to this analysis with WAVE, we have Sainsbury’s Supermarkets, Walmart, Target Corporation, Macy’s, IKEA, H&M Hennes, Chewy, Kroger, QVC, and Nike.

Figure 7 E-commerce websites evaluated.

The table shows the level of web accessibility of the 10 most accessible websites evaluated with the WAVE automatic review tool.

In addition, the correlation between the ranking of e-commerce sites and accessibility barriers was analyzed. In Table 4, the test statistic p > 0.05 for Errors, Contrast Errors and Ranking have a normal distribution despite their variability. While applying Lilliefors significance correction, the variables Errors and Contrast Errors p < 0.05 confirm that they do not have a normal distribution. However, the variable Ranking with Lilliefors significance correction has a p > 0.05, confirming a normal distribution.

Table 4 Normality tests.

Normality tests for Lilliefors significance correction. We applied for errors, contrast errors, and ranking.

Kolmogorov–Smirnov one-sample test	
	Errors	Contrast errors	Ranking	
N	50	50	50	
Normal parametersa,b	Media	24.94	36.08	25.50	
Standard deviation	34.064	71.184	14.577	
Maximum extreme differences	Absoluto	0.248	0.323	0.065	
Positivo	0.248	0.323	0.065	
Negativo	−0.232	−0.306	−0.065	
Test statistic=p	0.248	0.323	0.065	
Asymptotic significance (bilateral)	0.000c	0.000c	0.200c,d	
Notes:

a The test distribution is normal.

b It is calculated from data.

c Lilliefors significance correction.

d This is a lower limit of true significance.

Table 5 presents Spearman’s non-parametric correlation between e-commerce website ranking and accessibility barriers. In this case, the correlation is significant for accessibility barriers at the 0.05 level (bilateral).

Table 5 Spearman correlation.

Spearman’s non-parametric correlation between the ranking of e-commerce websites and accessibility barriers. Spearman’s Rho correlation is 0.329, indicating that the correlation is low positive.

	Ranking	Errors	
Spearman’s Rho	Ranking	Correlation coefficient	1	0.141	
Sig. (bilateral)		0.329	
N	50	50	
Errors	Correlation coefficient	0.141	1	
Sig. (bilateral)	0.329		
N	50	50	

In analyzing the accessibility of e-commerce websites, we applied multivariate descriptive statistics and pivot tables with the Excel tool. In addition, to analyze the correlation between the ranking of e-commerce websites and accessibility barriers, we applied the IBM SPSS Statistics version 25 statistical software with which we applied the Kolmogorov–Smirnov and Lilliefors significance correction. We found a non-parametric correlation, so we applied Spearman’s Rho between the ranking of e-commerce websites and the accessibility barriers of e-commerce websites. The correlation is 0.329, which indicates that the correlation is low positive.

Discussion

Concerning the modification made in WCAG-EM 1.0, three additional phases were included. By applying WAVE in phase 5, it is possible to perform automatic checks, which considerably reduces review time and detects numerous problems that would take more time and would be difficult to identify manually. Our methodology allows linking WCAG 2.1 criteria to accessibility barriers; this method can be applied throughout the website development cycle.

Currently, e-commerce sites have become essential tools to perform commercial transactions due to the COVID-19; with the findings obtained, it is evident that web designers and developers need to utilize the WCAG 2.1 (World Wide Web Consortium, 2018) to have more accessible and inclusive e-commerce sites. We identified that there is a significant number of e-commerce sites that occupy the top positions in the most developed countries such as the United States, United Kingdom, Greater China, Germany, Russia, and France; however, occupying the top positions do not guarantee that they comply with the WCAG 2.1 accessibility standards.

The most accessible store is Sainsbury’s Supermarkets, located in the United Kingdom with zero “Errors”, followed by Walmart, Target, Chewy, IKEA, and Macy’s, from the United States and H&M from Germany with one “Errors”, the rest of the e-commerce websites have more than one “Errors” and more than six “Contrast Errors”.

In the evaluation of e-commerce websites, it was found that the most common errors are related to “Contrast errors”, representing 54.4% of the total, in second place “Non-text Content”, corresponding to 25.6%, “Link Purpose” corresponds to 11.6% and “Info and Relationships” with 3.1% of the total.

Of the 50 e-commerce websites evaluated, 44.2% of the total comply with level “A” for accessibility and 55.8% with level “AA”; none of the sites evaluated reach level “AAA”.

The highest number of accessibility barriers are condensed in the “perceivable” principle, representing 83.1% of the total, while 16.9% are distributed among the “operable,” “robust,” and “understandable” principles. This finding implies that more barriers are related to problems for users with low vision, including older adults (Padmanaban, Konrad & Wetzstein, 2019); vision deteriorates with age due to the eye’s normal aging process. The existence of ocular degenerative diseases such as glaucoma, age-related macular degeneration, diabetic retinopathy, age-related cataracts, and cardiovascular accidents can mainly trigger a decrease in visual acuity and the visual field.

One of the most frequently repeated barriers on e-commerce sites is contrast and color usage, vital parameters for web accessibility. In contrast, most of the world’s users are visually impaired. According to the World Health Organization (2021), worldwide, at least 2.2 billion people have near or distance vision problems.

Achieving accessibility on e-commerce websites is an excellent challenge, so it is essential to apply the contrast ratio, which measures the difference in “luminance” or perceived brightness between two colors. The difference in brightness is expressed as a ratio varying from 1:1. WCAG 2.1 suggests addressing contrast with the three success criteria, 1.4.3 Contrast (minimum), 1.4.6 Contrast (enhanced) and 1.4.11 Contrast without text. In WCAG 2.1, it is suggested that 4.5:1 is the minimum required. Possibly some of these combinations are not very readable for all users.

For an e-commerce website to be accessible (World Wide Web Consortium, 2018), it must meet level AA for accessibility; the evaluated e-commerce websites do not present any accessibility statement or specify whether they officially comply. Since the home page of these e-commerce sites is the user’s entry point, it should be a primary objective for enhancements in terms of accessibility. Improving e-commerce websites should be a responsibility that companies take on, as it enhances the user experience and reduces the technology gap that people with disabilities may experience.

Accessibility issues can be reduced with the following recommendations (1) Improve contrast by considering the colors and contrasts of the screen, checking that they are displayed correctly on all devices. (2) Eliminate time limits, or at least lengthen them. It is essential to consider that users with disabilities need more time to browse online. (3) Always adding the option to “skip content” is very useful for users who access the Internet with screen readers and avoid content that does not interest them. (4) Provide transcripts of texts: in addition to subtitling videos, it is advisable to include transcripts so that the hearing impaired can read the video content at their own pace. (5) Add captions to graphics: especially those whose description does not conform to the “alt” attribute. The use of the “Alt” attribute, called “alternative text,” is essential for blind users who use screen readers. (6) Avoid using red to highlight important things; it would be fine if no people had color blindness problems. A good accessibility option is to use more giant letters or representative icons. (7) Write contents oriented to any reader, with and without disabilities. (8) Use legible fonts larger than 15px. (9) Avoid using paragraphs longer than four lines. (10) Use images and diagrams that help the reader understand the content better.

This study brings originality since the 50 top-ranked e-commerce sites were evaluated. Nowadays, e-commerce websites must have accessibility policies and standards; several transactions are made electronically due to the pandemic.

This research can guide developers and designers of e-commerce websites to spread the use of WCAG 2.1, which is intended to cover a more extensive set of recommendations to make the web more accessible. WCAG 2.1 can be considered as a superset containing WCAG 2.0. Therefore, as WCAG 2.1 extends WCAG 2.0, there are no incompatible requirements between one version.

Limitations

This research has a fundamental limitation; it was evaluated using the WAVE automated review tool. Despite being a powerful tool that helps organizations improve the accessibility of websites for people with disabilities, WAVE cannot tell whether web content is accessible; only a human being can determine true accessibility. This evaluation did not include testing with users with disabilities; three accessibility experts conducted accessibility testing. No additional hardware or digital ramps were used in this study to achieve greater accessibility during the website evaluation process.

Conclusions

We consider the research relevant, especially during the COVID-19 period, when e-commerce is considered the leading solution in confining and indirectly improving the global economy. The procedure for evaluating e-commerce websites with the WAVE tool can be applied to any website to make it more than accessible. We recommend performing evaluations with heuristic methods based on WCAG 2.1 (World Wide Web Consortium, 2018) accessibility barriers and automated reviews with users with different disabilities. We found that 55.8% of the websites reach the “AA” level suggested by WCAG 2.1. We found a low positive correlation between the rating of e-commerce websites and accessibility barriers according to Spearman’s Rho of 0.329. The study revealed that 25.6% of e-commerce websites present images of the products they offer and that 54.4% of the sites present contrast problems related to the perception principle, which need to be solved urgently to make the sites more inclusive. Finally, we suggest that business people, governments, and academia work in multidisciplinary teams to generate laws and regulations related to web accessibility that would benefit all users, especially those with disabilities.

Future Work

It is recommended to: (1) Perform tests with other automatic review tools and compare the results obtained for future research; (2) conduct tests with users with different disabilities; (3) build a software tool that includes artificial intelligence algorithms that help the software learn the heuristics that may cause accessibility barriers; (4) include hardware limitations and interstitial advertising in the study.

Additional Information and Declarations

Competing Interests

Author Contributions

Ethics

Data Availability

The authors declare that they have no competing interests.

Patricia Acosta-Vargas conceived and designed the experiments, performed the experiments, analyzed the data, performed the computation work, prepared figures and/or tables, authored or reviewed drafts of the paper, and approved the final draft.

Belén Salvador-Acosta performed the experiments, analyzed the data, performed the computation work, prepared figures and/or tables, authored or reviewed drafts of the paper, and approved the final draft.

Luis Salvador-Ullauri performed the experiments, analyzed the data, performed the computation work, authored or reviewed drafts of the paper, and approved the final draft.

Janio Jadán-Guerrero performed the experiments, performed the computation work, authored or reviewed drafts of the paper, and approved the final draft.

This research uses only publicly available information and no access to any other source besides the Mendeley repository dataset. This research constitutes a study of e-commerce websites (i) does not interact with any users; (ii) does not infer any new personal information; therefore, this study is exempt from an institutional review board. This research provides a data-driven audit to make better-informed decisions about accessibility on e-commerce websites, with the expected benefit of overcoming accessibility barriers faced by users of e-commerce websites in the future.

The following information was supplied regarding data availability:

The data is available at Mendeley: Acosta-Vargas, Patricia; Salvador-Acosta, Belén; Salvador-Ullauri, Luis; Gonzalez, Mario; Jadan-Guerrero, Janio (2021), “(Dataset) Web accessibility of the e-commerce website”, Mendeley Data, V1, DOI 10.17632/s33r57h5zm.1.

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
