# Peer review of "Accessibility challenges of e-commerce websites"

_PeerJ Computer Science, doi:10.7717/peerj-cs.891_

## Round 0.1 · original submission · Major Revisions

I agree with the two reviewers. The issues highlighted must be addressed in the revised version. I would like you to add tabular analysis of related work, and highlight your contributions.

Reviewer 1 ·

Basic reporting

This paper is well written. In addition to the introduction and conclusion, it is composed of method and material, results and discussion sections Hence, It is well structured.
The literature references are relevant and recent. However, almost of theme are reports and guides from international organizations like WHO. In fact, it will be more suitable to cite academic papers.
The authors follow a clear methodology. The obtained results are analyzed and discussed in order to give solution to the accessibility problem.
Solutions of accessibility are described in the conclusion section. I think that it is more suitable to put in the discussion section. Also, I think that the authors have to describe in details the followed methodology and the proposed solutions.
The authors have to explain some abbreviations (like ARIA attributes, A, AA and AAA levels).
I think that the introduction is very long. The authors touch on many subjects to show the importance of the problem (like COVID-19, e-commerce, accessibility). I suggest to rewrite the introduction to be more consistent.
I remark that there is no related work section presented in this paper !!!

Experimental design

The paper presents the accessibility challenges of e-commerce websites. The authors through this study have presented many contributions: 1 - correlation between the ranking of e-commerce websites and the accessibility, 2 - the most significant accessibility barriers and neglected principles 3 - some proposed solutions to improve the accessibility of e-commerce websites. I think that the research is well designed. Also, the research question is relevant, especially during COVID-19 period where e-commerce is considered as a main solution to apply confinement.
The applied methodology is rigorous. Also, the authors have mentioned the main limits of this methodology which consists mainly in the inability of automated tools to test content accessibility.
I think, that the authors have to explain in details the followed methodology.
Comparing to some cited works (like Oliveira et al (2020)), , I did understand exactly the novelty of this study. Is it the proposition of solutions to the accessibility? because the same tools (WAVE) and field (E-commerce) are used in both studies.

Validity of the findings

The obtained results are very important and they allow improving e-commerce websites (and indirectly improving the global economy). The first remark of the authors is the correlation between e-commerce websites ranking and the accessibility. This remark guides the business sector to improve the accessibility of their websites to improve their rank. Also, the authors describe the main accessibility barriers and the most neglected principles of this attribute. Moreover, the authors presented the solutions of this issue.
In term of novelty and originality, the authors have mentioned several recent works that are deal with this problem (using the same tools and applied in the e-commerce field). I think, that the authors have to mention clearly the novelty and originality points comparing to these studies. I suggest adding related work section to study this point.
The authors provided all data required to replicate.
In future works, the authors suggest using in their future work a hybrid method by applying automated tool with manual review method. I think that it is difficult to test the accessibility of 50 websites manually. Also, the manual method causes subjectivity problems.

Reviewer 2 ·

Basic reporting

The manuscript titled: Accessibility challenges of e-commerce websites, evaluates the accessibility of 50 e-commerce web sites in the top rankings based on classification proposed by ecommerceDB. The evaluation was made under WAVE tool. MS Excel pivot table tool was applied for analyzing the accessibility of selected e-commerce. In constrast, the normality test and the correlation between the ranking of e-commerce websites and accessibility barriers were analyzed using IBM SPSS (v 25).
Overall, the authors’ contribution was not pretty clear since the study analyses the outputs of WAVE tool on 50 top ranking e-commerce web sites. The proposed Methodology for evaluating e-commerce websites needs further highlights. For exemple, in the third phase: Define the test scenario, from a software engineering point of view there is no provided test. scenario. Also, Is there any potential back-tracking in the proposed Methodology in the case where WAVE does not exhibit a relevant information (potential failures, new accessibility improvements purposes, etc) ? Also, in Phase 2, which user categories were selected? it should be discussed.

The authors are strongly invited to make a comparison between their findings and the study of (Xu, 2020) which is the unique work presented in the State-Of-The-Art section that uses WAVE-based evaluation of 45 e-commerce website.
I should be interesting, for authors, to highlight the WAVE analysis method and the additional modification indicated in WCAG-EM 1.0 since it impacts the results and the proposed accessibility improvements.
In table 2 the column headings were not explained (Errors, Contrast Errors, Alerts, Features, Structural Elements and ARIA). For instance, the authors indicate errors and contrast errors, what is the difference between them? Also, it seems that ARIA, Structural Elements, Features and Alerts are warning barrier types. I suggest to explain these concepts when introducing table2..
Table 3 is presented in summative manner and needs more clarification about Success criteria and Level. Success criteria presents number with X.X.X format which is not clear (the meaning of the first, second and third number is missing). Also, level takes A or AA values which is even not explained before table3. In short, success criteria according to WCAG 2.1 should be outlined first in the appropriate position in the paper.


The authors should mention how Accessibility barriers have been identified? if the identification process belongs to WAVE findings, it must be noted.
In table 4, the authors are invited to justify why the Normality testing is required? Also, the result of normality test should be commented either the data follow the normal distribution or not (i.e., Lilliefors test)
In the discussion section, in addition to what is raised about accessibility barriers (low vision, ocular degenerative diseases, cardiovascular accidents, etc), the Hardware limitations were not discussed. Sometimes end-users’ devices (smartphone, tablets, …) may hide some significant information on the web site based on automatic brightness functionality which is proposed mainly for preserving human vision. Interstitial advertising was not discussed as well.
In table 5, when calculating the Pearson correlation, the authors should provide at which level the correlation is significant. Using SPSS, the significance level is defaulted by 0.01, it there any modification of that level?
Page 11, line 282, (In analyzing the accessibility of e-commerce websites, we applied descriptive statistics …)  the authors should clarify which type of descriptive statistics is used: univariate or multi-variate
Finally, as a native question: Is it possible to take into account web applications by WAVE-based evaluation?
Although figures, tables are well labelled and described, some minor issues are:
• Line 94 the title: The state of the art is missing since the what follows discuss literature review.
• Second paragraph after the figure 1: (there are many; many people with hearing roblems)-> redundance
• Table3, figure 5, figure 6 and figure 7 captions are written in bold. It should be submitted to the template.
• (…, a sample of the top 50 e-commerce websites was taken from the EcommerceDB ranking site)  recurrent phrase in the manuscript. (In Introduction section, In Materials & Methods section, in results section)
• Page 8, line 125, (Our research proposes an automatic review method using the WAVE Web Accessibility Evaluation Tool (15) to solve …)  what (15) stands for? Is it about the tool version?
• page 7, ligne 85: WCAG 2.1 (World Wide Web Consortium, 2018) consists of 4 principles, 13 guidelines and conformance criteria, plus an undetermined number of suitable techniques  Is there really an undetermined number? Can it be limited?
• Page 7, line 88: Principle 1, related to perceptibility, refers to information, and user interface components should be presented most simply. Principle 2 focuses on operability - it comprises the user interface components, and navigation should be between each page  the difference between principle 1 and 2 is not clear.
• The first paragraph in discussion section, (it was revealed that there is a need for web developers and web designers to apply WCAG 2.1 …) : incomplete phrase, a descriptive word is missing after WCAG 2.1 like : guidelines, principles, requirements.
• Page 13, line 340, (This research can guide developers and designers of e-commerce websites to spread the use of WCAG 2.1 (6), ..) what does (6) stands for ?

Experimental design

Rigorous investigation is performed in a simplest way. Overall, the authors’ contribution was not pretty clear since the study analyses the outputs of WAVE tool on 50 top ranking e-commerce web sites. The proposed Methodology for evaluating e-commerce websites needs further highlights. The introduction should also highlights what are the gaps left by existing solutions and how the presented proposal addresses them.

Validity of the findings

In conclusion section, the provided recommendations are suitable for any type of web sites. I would have liked that they are dedicated to e-commerce sites since the core matter of the study is the Accessibility issue in e-commerce websites. Generally, e-commerce websites present images of products. I suppose that an image processing issue should be tackled.
The authors suggest future works in two places in the manuscript: before and within the conclusion section. It should be better to combine suggested future works in the end of conclusion section.

---

## Round 0.2 · Minor Revisions

We thank you incorporating reviewer comments. Before we proceed further, I feel that there are some more changes that are required to be made. Therefore, you should upload the modified version and thereafter we will further process your paper.

Reviewer 2 ·

Basic reporting

Overall, the raised comments are handled.

Paper formatting needs more adjustments. As well as English improvement.

For example, in Page 10, line 211 : (This phase starts the evaluation process is defined, the level of compliance with WCAG 2.1 for the evaluation is defined (World Wide Web Consortium, 2018). Should be reformulated.

In Materials & Mathods section, page 195, (We used (WebAIM, 2021) Web Accessibility Evaluation Tool (WAVE) ) -> the citation should be putted after the tool name.

Experimental design

The authors are invited to justify why they put the works ((Villa & Monzón, 2021), (Pollák et al., 2021) and (Paștiu et al., 2020)) to denote the impact of COVID-19 on the growth of e-commerce websites? Is there a need for making that? Also, the main issue of the paper is the accessibility not the impact of COVID-19 on the evolution of e-commerce websites. I suggest to make just a short paragraph in the form: according to ((Villa & Monzón, 2021), (Pollák et al., 2021) and (Paștiu et al., 2020)) the COVID-19 impacts the growth of e-commerce Websites in the way …..) not to describe each work separately.

In the Literature review of COVID-19, e-commerce, and web accessibility section, I suggest to remove (COVID-19, e-commerce, and web accessibility) and to keep only the title: Literature review
From the line 103 to 136, the authors expose the WCAG 2.1 principles in the literature review section after introducing COVID-19 works related to Web accessibility. I suggest to put a brief description of WCAG principles in the introduction section and restructure it better or to make a separated section that outlined WCAG principles.

The paper contribution is well designed.

Validity of the findings

The previous remarks and suggestions have been reviewed.

---

## Round 0.3 · Minor Revisions

I now have the reviewer's report. I have analysed the report and have reviewed your submission again. I now feel that paper is almost ready to be accepted, but there are just a few more modifications required.

Reviewer 2 ·

Basic reporting

I have just one last remark regarding the order of sections: it should be better to put Web accessibility principles section (from line 92 to 125) after the literature review section and before Materials & Methods section since it’s inappropriate in scientific writing to make sections intertwined. I think that my suggestion in the last review was not clearly understood.

For the rest, I’m satisfied and I think that the raised comments have been handled.

Experimental design

/

Validity of the findings

/

Additional comments

/

---

## Round 0.4 · accepted · Accept

I am happy to see that after the 3 rounds of reviews, your paper is now in perfect form to be published. Congratulations and thanks for submitting to PeerJ Computer Science.